# Integrin α2β1 Represents a Prognostic and Predictive Biomarker in Primary Ovarian Cancer

**DOI:** 10.3390/biomedicines9030289

**Published:** 2021-03-12

**Authors:** Katharina Dötzer, Friederike Schlüter, Franz Edler von Koch, Christine E. Brambs, Sabine Anthuber, Sergio Frangini, Bastian Czogalla, Alexander Burges, Jens Werner, Sven Mahner, Barbara Mayer

**Affiliations:** 1Department of General, Visceral and Transplant Surgery, University Hospital, Ludwig-Maximilians-University Munich, Marchioninistraße 15, 81377 Munich, Germany; katharina.doetzer@med.uni-muenchen.de (K.D.); friederike1.schlueter@med.uni-muenchen.de (F.S.); jens.werner@med.uni-muenchen.de (J.W.); 2Gynecology and Obstetrics Clinic, Klinikum Dritter Orden, Menzinger Straße 44, 80638 Munich, Germany; franz.koch@dritter-orden.de; 3Department of Obstetrics and Gynecology, Klinikum Rechts der Isar, Technical University Munich, Ismaninger Straße 22, 81675 Munich, Germany; christine.brambs@tum.de or; 4Department of Obstetrics and Gynecology, Starnberg Hospital, Oßwaldstraße 1, 82319 Starnberg, Germany; sabine.anthuber@klinikum-starnberg.de; 5Department of Obstetrics and Gynecology, Munich Clinic Harlaching, Sanatoriumsplatz 2, 81545 Munich, Germany; sergio.frangini@muenchen-klinik.de; 6Department of Obstetrics and Gynecology, University Hospital, Ludwig-Maximilians-University Munich, Marchioninistraße 15, 81377 Munich, Germany; bastian.czogalla@med.uni-muenchen.de (B.C.); alexander.burges@med.uni-muenchen.de (A.B.); sven.mahner@med.uni-muenchen.de (S.M.); 7German Cancer Consortium (DKTK), Partner Site Munich, Pettenkoferstraße 8a, 80336 Munich, Germany

**Keywords:** primary ovarian cancer, integrin α2β1, prognostic factor, predictive factor, immune infiltrate, targeted therapy, personalized medicine

## Abstract

Currently, the same first-line chemotherapy is administered to almost all patients suffering from primary ovarian cancer. The high recurrence rate emphasizes the need for precise drug treatment in primary ovarian cancer. Being crucial in ovarian cancer progression and chemotherapeutic resistance, integrins became promising therapeutic targets. To evaluate its prognostic and predictive value, in the present study, the expression of integrin α2β1 was analyzed immunohistochemically and correlated with the survival data and other therapy-relevant biomarkers. The significant correlation of a high α2β1-expression with the estrogen receptor alpha (ERα; *p* = 0.035) and epithelial growth factor receptor (EGFR; *p* = 0.027) was observed. In addition, high α2β1-expression was significantly associated with a low number of tumor-infiltrating immune cells (CD3 intratumoral, *p* = 0.017; CD3 stromal, *p* = 0.035; PD-1 intratumoral, *p* = 0.002; PD-1 stromal, *p* = 0.049) and the lack of PD-L1 expression (*p* = 0.005). In Kaplan–Meier survival analysis, patients with a high expression of integrin α2β1 revealed a significant shorter progression-free survival (PFS, *p* = 0.035) and platinum-free interval (PFI, *p* = 0.034). In the multivariate Cox regression analysis, integrin α2β1 was confirmed as an independent prognostic factor for both PFS (*p* = 0.021) and PFI (*p* = 0.020). Dual expression of integrin α2β1 and the hepatocyte growth factor receptor (HGFR; PFS/PFI, *p* = 0.004) and CD44v6 (PFS, *p* = 0.000; PFI, *p* = 0.001; overall survival [OS], *p* = 0.025) impaired survival. Integrin α2β1 was established as a prognostic and predictive marker in primary ovarian cancer with the potential to stratify patients for chemotherapy and immunotherapy, and to design new targeted treatment strategies.

## 1. Introduction

Several clinicopathological factors, such as advanced tumor stage and residual tumor after surgery, have been established as strong prognostic factors in primary ovarian cancer [1]. In addition, a few tumor biological characteristics have been identified as prognostic markers. Examples are distinct gene signatures [2] or a high number of T-cells [3]. Although recently new promising candidates were detected [4], predictive markers are rare in primary ovarian cancer. Two targeted therapy approaches are recommended under current guidelines, namely vascular endothelial growth factor (VEGF) inhibition and poly (ADP-ribose) polymerase (PARP) inhibition, which are both administered in addition to the standard chemotherapy [5,6,7]. For VEGF inhibition, no predictive biomarker is available to select appropriate patients for anti-angiogenic therapy. Similarly, BRCA mutation or HRD status, which so far represent a prerequisite for some of the PARP inhibition treatments, need to be re-evaluated [8]. Thus, robust biomarkers for precise prognosis and treatment response are urgently required in primary ovarian cancer. This importance is emphasized by the fact that, despite standard therapy combining radical surgery and adjuvant platinum-based chemotherapy, 70–80% of the patients suffer from relapse [9].

Integrins are transmembrane cell adhesion molecules, which mediate cell–cell and cell–extracellular matrix (ECM) interaction. Currently, 18 α-subunits and 8 β-subunits are identified, forming a variety of integrin heterodimers [10]. Due to their ability of inside-out and outside-in signaling, they are known to be involved in migration, invasion, and metastasis promoting tumor progression in several cancer types [11,12,13]. Considering the mechanism of ovarian cancer metastasis by spreading in the peritoneal fluid and attaching to the omental and peritoneal tissue [14,15,16], integrins seem to be a promising therapeutic target in ovarian cancer.

While there is already some information about ovarian cancer and other β1-heterodimers, such as integrins α4β1 and α5β1 [17], less information is available about integrin α2β1. The main ligand of integrin α2β1 is collagen type I, but binding to other collagen types, laminins, and other ECM-proteins is also possible [18,19]. Expression of integrin α2β1 is not only observed on the epithelial cells, but also on the endothelial cells, platelets, white blood cells, and fibroblasts [20,21].

Previous studies indicate a role of integrin α2β1 in chemotherapy resistance [22,23], which constitutes a special interest for ovarian cancer. In the present study, the expression of integrin α2β1 in primary ovarian cancer and its prognostic and predictive role will be evaluated.

## 2. Materials and Methods

### 2.1. Study Population

Forty-eight patients diagnosed with a primary, chemonaive ovarian, fallopian tube, or peritoneal cancer from the SpheroID-Study were included. Patients suffering from another neoplasia within the last five years were excluded. Patients were recruited between September 2012 and January 2015 from five ovarian cancer centers, namely University Hospital, LMU Munich (n = 16), Klinikum Dritter Orden (n = 15), Klinikum rechts der Isar, Technical University Munich (n = 7), Munich clinic Harlaching (n = 5), and Starnberg Hospital (n = 5). Standardized surgical resection and pathological analysis was conducted by the recruiting hospital. Patient-, tumor- and treatment-related data for correlations were given in the routine reports and delivered in a pseudonymized form. Survival analysis was performed after the completion of chemotherapy. Seven patients with no chemotherapy or a reduced number of treatment cycles (≤ 2) had to be excluded. Progression-free survival (PFS) was defined as the time from surgical treatment to relapse or progression. Platinum-free interval (PFI) was defined as the time from end of the chemotherapy to relapse or progression. Overall survival (OS) was defined as the time from surgical treatment to death. Data from patients who did not die and had no relapse or progression were censored at the date of their last visit.

### 2.2. Immunohistochemistry

After surgical removal, tumor samples were snap frozen in liquid nitrogen. Serial cryosections (5 µm) were performed. The samples were stained immunohistochemically using the avidin–biotin–peroxidase method [24]. Tissue sections were fixed either in acetone for 8 min or, for the antigens ERα and PgR, in formalin for 3 min and afterwards in a citrate buffer for 7 min at 90 °C. Blocking of unspecific Fc receptors was performed with 10% AB Serum (Biotest, Dreieich, Germany) in either PBS (acetone fixation) or in a TRIS–HCl buffer (formalin fixation) for 20 min. Endogenous biotin was blocked with a two-step avidin–biotin blocking kit (Vector Laboratories, Burlingame, CA, USA) according to the manufacturer’s instructions for 20 min. Primary antibodies were applied for one hour. Details about primary and secondary antibodies and working concentrations, including the appropriate positive and negative controls, are given in Table 1. Secondary biotinylated antibodies and peroxidase conjugated streptavidin (Dianova, Hamburg, Germany) were incubated for 30 min each.

### 2.3. Evaluation of Biomarker Expression

Sections were evaluated semiquantitatively using a light microscope (Appendix A). The percentage of positively stained carcinoma cells was evaluated for each antigen. Tumors were defined as hormone receptor-positive if ≥1% of the cancer cells revealed a nuclear staining of ER or PR [25]. Her2/neu expression was scored according to breast cancer [26] and gastric cancer [27] guidelines. Due to the lack of further references, the other biomarkers’ expression was estimated as a percentage of positive cancer cells in 10% steps. Validation was conducted by a second observer (FS). In the absence of standardized cut-offs for other biomarkers, cut-offs were evaluated according to the biphasic distribution or the group size (see Table 1). Quantitative evaluation of CD3, CD8, and PD-1, and semiquantitative evaluation of PD-L1 was performed according to Dotzer et al. [24].

### 2.4. Statistical Analysis

Clinicopathological factors were grouped by clinical relevance. Integrin expression was correlated with clinicopathological factors, other biomarkers’ expression, and immune infiltrate using the Fisher’s exact two-tailed test. Univariate analysis was performed by calculating cumulative survival probabilities with the Kaplan–Meier method and comparing them with a log-rank test. A Cox regression model was used for the multivariate analysis of survival. *p*-values < 0.05 were considered to be statistically significant. All statistical analyses were performed using IBM SPSS Statistics 23 (Armonk, NY, USA).

## 3. Results

### 3.1. Patient Characteristic

The clinicopathological data are shown in Table 2. Forty-eight patients were included in this study. The mean age at time of diagnosis was 62 years. Most patients suffered from high-grade, serous ovarian carcinoma in an advanced FIGO (Fédération Internationale de Gynécologie et d’Obstétrique) stage with the presence of ascites. Complete surgical resection without macroscopic residual tumor was achieved in 72.9% of all patients. In total, 83.4% of the patients received chemotherapy based on carboplatin and paclitaxel. The median OS was 42 months, the median PFS was 22 months, and the median PFI was 17 months.

Survival data are summarized in Table 2. The presence of distant metastases (FIGO IV) was related to a shorter OS (*p* = 0.015) and tended to predict a shorter PFS (*p* = 0.081) and PFI (*p* = 0.068). Furthermore, patients with a macroscopic residual tumor after surgery showed a significant shorter OS (*p* = 0.041), PFS (*p* = 0.008) and PFI (*p* = 0.01).

### 3.2. Prognostic and Predictive Impact of Integrin α2β1

High integrin α2β1 expression in primary ovarian cancer was found to be associated with an unfavorable prognosis. Patients with a high expression of integrin α2β1 showed a median PFS of 16 months, which was significantly shorter compared to patients with low α2β1 expression (PFS 29 months, *p* = 0.035). In addition, high expression of integrin α2β1 predicted a shorter PFI (11 months) in contrast to patients with a low α2β1-expressing primary tumor (25 months, *p* = 0.034). Most importantly, a high expression of integrin α2β1 in primary ovarian cancer was found to be an independent prognostic factor for a shorter PFS (HR 2.46, CI 95% 1.14–5.29, *p* = 0.021) and a shorter PFI (HR 2.44, CI 95% 1.14–5.26, *p* = 0.022). No impact of the extent of α2β1 expression on OS was observed (Table 3). In addition, no significant correlation between the expression of integrin α2β1 and clinicopathological factors could be found.

### 3.3. Correlation of Integrin α2β1 with Other Biomarkers

In almost all patients (17 out of 18, 94.4%), a high expression of integrin α2β1 significantly correlated with a high expression of ERα (*p* = 0.035). Furthermore, a high expression of integrin α2β1 could be found more frequently in patients with a high expression of EGFR (7 out of 10, 70%) compared to patients with a low expression of EGFR (11 out of 38, 28.9%, *p* = 0.027, Table 4).

### 3.4. Prognostic and Predictive Impact of Integrin α2β1 Combined with Other Biomarkers

The dual expression of integrin α2β1 and various growth factor receptors revealed an impact on PFS and PFI (Table 5). Patients with a high expression of integrin α2β1 and a positive Her-2/neu status showed a shorter PFS (*p* = 0.043) and PFI (*p* = 0.037) than patients with a low expression of integrin α2β1, Her-2/neu, or both. Combined high expression of integrin α2β1 and IGF1R correlated significantly with a shorter PFS (*p* = 0.045) and PFI (*p* = 0.043). Most interestingly, a high expression of integrin α2β1 and HGFR was related to a shorter PFS (*p* = 0.004) and PFI (*p* = 0.004) and impaired prognosis in comparison to integrin α2β1 as single biomarker.

Likewise, a high expression of both integrin α2β1 and CD44v6 was found to be a strong factor in a poor prognosis that correlated with a shorter PFS (*p* = 0.000), PFI (*p* = 0.001) and a reduced OS (*p* = 0.025, Table 5).

### 3.5. Correlation of Integrin α2β1 and Immune Infiltrate

In patients with a high expression of integrin α2β1, low numbers of stromal and intratumoral CD3+ cells were found (14 out of 18, 77.8%, *p* = 0.035 and *p* = 0.017, Table 6). Furthermore, most tumors with a high expression of integrin α2β1 showed a low density of stromal (16 out of 18, 88.9%, *p* = 0.049) and intratumoral (17 out of 18, 94.4%, *p* = 0.002) PD-1+ cells. PD-L1 positivity was found more often in tumors with a low expression of integrin α2β1 (23 out of 30, 76.7%) compared to samples with a high expression (6 out of 18, 33.3%; *p* = 0.005). No correlations for CD8+ cells have been found.

## 4. Discussion

In the present study, integrin α2β1 was identified as a potential new prognostic and predictive marker in primary ovarian cancer.

A high expression of integrin α2β1 was identified as a marker for a poor prognosis with equal strength, as reported for the established clinical factors: FIGO stage and macroscopic residual tumor after surgical resection. The positive correlation between a high expression of the integrin β1 chain and short survival is documented for various tumor entities [28,29,30]. In particular, integrin α5β1 is already known to be an unfavorable prognostic factor for ovarian cancer [31], but also for cervical, gastric, and non-small-cell lung cancer [32,33,34].

Integrin α2β1 is involved in many steps of cancer progression. Binding to components of the extracellular matrix (ECM), integrin α2β1 mediates tumor cell invasion and metastasis [35,36,37]. This step is promoted by crosstalk with growth factor receptors [38,39]. Interestingly, in the present study, a combined expression of integrin α2β1 with ERα and EGFR was observed. Furthermore, the signaling of integrin α2β1 can induce chemoresistance. This mechanism was observed for chemotherapies containing paclitaxel [23,40], gemcitabine [41], and etoposide [42].

Early relapse and resistance to platinum-based chemotherapy are key problems in the treatment of ovarian cancer [43]. Therefore, the predictive value for the treatment response of integrin α2β1 was analyzed in the present study. Patients with a high expression of integrin α2β1 were observed to have a shorter median PFI. In particular, β1 integrins are already known to promote platinum resistance in ovarian cancer. The mechanisms of this effect are still unclear. Intracellular signaling initiated by binding to the ECM seems to be fundamental for cell adhesion-mediated drug resistance (CAM-DR) [44,45]. One of the main ECM molecules involved in this concept is collagen type I [46], which is the central binding partner of integrin α2β1 [18]. These molecular interactions suggest that the heterodimer α2β1 contributes to CAM-DR. Therefore, targeting integrin α2β1 represents a promising strategy for overcoming platinum resistance in primary ovarian cancer.

In addition, a high expression of integrin α2β1 was observed in patients with a low density of stromal and intratumoral CD3+ as well as PD-1+ cells. Inversely, more than 75% of patients with a low expression of integrin α2β1 showed PD-L1 positivity, which represents an established predictive biomarker for immunotherapy [47]. Several integrins are related to an immunosuppressive tumor microenvironment [48,49]. For example, αv-integrins are major activators of latent TGF-β, which is involved in immunotherapy resistance [50]. The present data suggest that integrin α2β1 might play a similar role. Recently, immunotherapy became a promising approach in ovarian cancer [51,52], and phase III studies with checkpoint inhibitors in combination with platinum-based chemotherapy are already ongoing (NCT03038100, NCT03740165, NCT03737643). Low expression of integrin α2β1, therefore, could be a potential predictive marker for immunotherapy in ovarian cancer. Taken together, integrin α2β1 represents a stratification marker for patients receiving platinum-based chemotherapy and immunotherapy.

Inhibition of integrin α2β1 should be considered as a targeted therapy in ovarian cancer. Several molecules and antibodies have been developed and evaluated for integrin α2β1 inhibition in other entities.

Anti-tumoral activity was shown in prostate cancer in vivo using the monoclonal antibody GBR-500 [53]. E-7820 is a sulphonamide derivative that inhibits the expression of α2-mRNA. In Phase I studies, treatment was associated with a stable disease in a variety of malignancies [54,55]. Phase II studies are ongoing to evaluate the combination with chemotherapy in colon carcinoma (NCT01347645, NCT01133990, NCT00309179). Another β1-antibody could improve the efficiency of platinum-based chemotherapy in non-small-cell lung cancer [56]. However, despite these promising approaches, the complex biology of heterodimers with promiscuous ligands, allosteric activation, and multiple intracellular signaling pathways might hinder successful treatment strategies [13,57,58].

Furthermore, the results of this study also indicate the potential efficiency of dual inhibition. Patients with a combined high expression of integrin α2β1 and HGFR or CD44v6 showed a very short median PFS and PFI, indicating a worse prognosis and platinum resistance.

Dual targeting has become a promising strategy in ovarian cancer. Its efficiency was proven in tumor spheroid and mouse models [59,60]; thus, various phase I studies are ongoing (NCT03895788, NCT03695380, NCT04315233). In future studies, dual inhibition including integrin α2β1-antagonists should be considered for patients with an appropriate biomarker profile.

The main limitation of this study is the small cohort, though it is representative and comparable to cohorts of other clinical trials. The promising role of integrin α2β1 as a new prognostic and predictive biomarker in primary ovarian cancer needs to be confirmed by an enlarged study.

## 5. Conclusions

In the present study, integrin α2β1 was identified as a prognostic and predictive marker in primary ovarian cancer. High expression of integrin α2β1 correlated with a short PFS. Prognosis was even worse in integrin α2β1-positive tumors co-expressing HGFR or CD44v6. This finding might lead to new biomarker-directed treatment strategies in primary ovarian cancer. In addition, the high expression of integrin α2β1 correlated with a short PFI, supporting the hypothesis that integrins mediate platinum resistance. Thus, a high expression of integrin α2β1 might represent a stratification marker for personalized treatment.

## Figures and Tables

**Table 1 biomedicines-09-00289-t001:** Biomarkers and antibodies.

Antigen	Clone	Species	Fixation	Use of Kit	wc (μg/mL)	Supplier	Cut-Off for Positivity
**Primary antibodies**							
Integrin α2β1	BHA2.1	m	Acetone	-	2.50	Millipore, Burlington, MA, USA	≥20%
ERα	1D5	m	Formalin	+	2.50	Dako, Santa Clara, CA, USA	≥1%
PR	PgR 636	m	Formalin	+	2.50	Dako, Santa Clara, CA, USA	≥1%
HER-2/neu	4B5	r	Acetone	-	1.50	Ventana, Roche, Basel, CH	≥10% (Intensity 2+/3+)
EGFR	H11	m	Acetone	-	2.94	Dako, Santa Clara, CA, USA	≥50%
HGFR	SP44	r	Acetone	-	2.12	Spring Bioscience, Pleasanton, CA, USA	≥50%
IGF1R	23-41	m	Acetone	+	4.00	invitrogen, Carlsbad, CA, USA	≥80%
MUC-1	Ma552	m	Acetone	-	0.50	Monosan, Uden, NL	≥70%
CD44v6	VFF-18	m	Acetone	-	1.00	affymetrix eBioscience, Santa Clara, CA, USA	≥10%
Integrin αVβ3	LM609	m	Acetone	-	5.00	Millipore, Burlington, MA, USA	≥20%
CD3	UCHT1	m	Acetone	-	1.25	BD Biosciences, Franklin Lakes, NJ, USA	
CD8	C8/144B	m	Acetone	+	3.00	Dako, Santa Clara, CA, USA	
PD-1	MIH4	m	Acetone	+	10.00	affymetrix eBioscience, Santa Clara, CA, USA	
PD-L1	MIH1	m	Acetone	+	10.00	affymetrix eBioscience, Santa Clara, CA, USA	≥1%
**Positive controls**							
Epithelial Antigen	Ber-EP4	m	Acetone	-	2.50	Dako, Santa Clara, CA, USA	
CD45	2B11 + PD7/26	m	Acetone	-	4.50	Dako, Santa Clara, CA, USA	
**Isotype controls**							
MOPC 21	MOPC 21	m		-	5.00	Sigma-Aldrich, St. Louis, MO, USA	
	MOPC 21	m		+	4.00	Sigma-Aldrich, St. Louis, MO, USA	
	MOPC 21	m		+	10.00	Sigma-Aldrich, St. Louis, MO, USA	
DA1E	DA1E	r		-	2.12	Cell Signaling, Danvers, MA, USA	
**Biotin conjugated secondary antibodies**							
	111-065-114	g anti r			7.00	Jackson Immunoresearch, West Grove, PA, USA	
	315-065-048	r anti m			0.75	Jackson Immunoresearch, West Grove, PA, USA	

**Legend:** wc: working concentration, m: mouse, r: rabbit, g: goat, all used antibodies’ isotype was IgG1. ERα: estrogen receptor α, PR: progesterone receptor, Her-2/neu: human epidermal growth factor receptor 2, EGFR: epidermal growth factor receptor, HGFR: hepatocyte growth factor receptor, IGF1R: insulin-like growth factor 1, MUC-1: mucin-1, PD-1: programmed cell death protein 1, PD-L1: programmed cell death-ligand 1.

**Table 2 biomedicines-09-00289-t002:** Patient characteristics.

		n or Value	%
**Age**	mean/median	62/66 years	
	range	24–83 years	
**FIGO Stage**	I or II	0	0.0%
	III	34	70.8%
	IV	14	29.2%
**pT**	pT2	5	10.4%
	pT3	43	89.6%
**pN**	pN0	6	12.5%
	pN1	31	64.6%
	Nx	11	22.9%
**cM**	cM0	34	70.8%
	cM1	14	29.2%
**Primary Tumor Site**	Ovarian	39	81.3%
	Fallopian Tube	6	12.5%
	Peritoneal	3	6.3%
**Histological Subtype**	Serous	44	91.7%
	Other	4	8.4%
**Grading**	G1/G2	2	4.2%
	G3	46	95.8%
**Ascites**	yes	40	83.3%
	no	8	16.7%
**Macroscopic Residual Tumor after Surgery**	None	35	72.9%
	<1 cm	6	12.5%
	>1 cm	7	14.6%
**First-Line-Treatment**	C	4	8.3%
	C + P	15	31.3%
	C + P + B	25	52.1%
	None	4	8.3%
**Relapse after Chemotherapy**	<6 months	2	4.2%
	6–12 months	12	25.0%
	>12 months	28	58.3%
	none or non-sufficient chemotherapy	6	12.5%

**Legend:** n: number of patients, Nx: no evaluation of lymph node status, C: carboplatin, P: paclitaxel, B: bevacizumab.

**Table 3 biomedicines-09-00289-t003:** Univariate and multivariate survival analysis of clinicopathological factors and integrin α2β1.

		PFS				PFI				OS	
	n	Log-Rank		MV Cox Regression		Log-Rank		MV Cox Regression		Log-Rank	
		MS	*p*	HR (CI 95%)	*p*	MS	*p*	HR (CI 95%)	*p*	MS	*p*
**Age ≤ 62 years**	19	22	0.965			17	0.970			nr	0.193
**Age > 62 years**	23	22				17				42	
**<pT3c**	7	27	0.665			22	0.679			45	0.928
**pT3c**	35	22				17				42	
**pN0**	5	29	0.163			17	0.145			45	0.929
**pN1**	28	22				22				42	
**cM0**	29	27				22				nr	
**cM1**	13	16	**0.081**	2.06 (0.92–4.62)	0.081	11	**0.068**	2.10 (0.94–4.69)	**0.072**	30	**0.015**
**G1/G2**	2	14	0.579			8	0.610			30	0.843
**G3**	40	22				17				42	
**Ascites absent**	6	35	0.147			30	0.139			42	0.408
**Ascites present**	36	19				15				38	
**MR Tumor absent**	30	27				22				45	
**MR Tumor present**	12	13	**0.008**	2.19 (1.03–4.68)	**0.043**	9	**0.010**	2.10 (0.99-4.51)	0.057	26	**0.041**
**Integrin α2β1 low**	27	29				25				45	
**Integrin α2β1 high**	15	16	**0.035**	2.46 (1.14–5.29)	**0.021**	11	**0.034**	2.45 (1.14-5.26)	**0.022**	30	0.155

**Legend:** n: number of patients, Cox regression: multivariate Cox regression, MS: median survival (in months) in Kaplan–Meier estimator, HR: hazard ratio, CI: confidence interval, MR Tumor: macroscopic residual tumor; nr: median survival not reached.

**Table 4 biomedicines-09-00289-t004:** Correlation between integrin α2β1 and other biomarkers.

				Integrin α2β1		
			n	<20%	≥20%	*p* ^#^
**Growth Factor-Receptor**	**ERα**		48			**0.035**
		**<1%**		10	1	
		**≥1%**		20	17	
	**PR**		48			0.127
		**<1%**		22	9	
		**≥1%**		8	9	
	**Her-2/neu**		48			1
		**negative**		22	13	
		**positive**		8	5	
	**EGFR**		48			**0.027**
		**<50%**		27	11	
		**≥50%**		3	7	
	**HGFR**		48			0.133
		**<50%**		16	5	
		**≥50%**		14	13	
	**IGF1R**		48			0.451
		**<80%**		4	4	
		**≥80%**		26	14	
**Cell-Adhesion-Molecule**	**MUC-1**		48			0.765
		**<70%**		14	7	
		**≥70%**		16	11	
	**CD44v6**		48			0.103
		**<10%**		24	10	
		**≥10%**		6	8	
	**Integrin αvβ3**		48			0.19
		**<20%**		24	11	
		**≥20%**		6	7	

**Legend:** n: number of patients, ^#^: *p*-value calculated by Fisher’s exact two-tailed test.

**Table 5 biomedicines-09-00289-t005:** Univariate survival analysis of dual expression of integrin α2β1 and other biomarkers.

		PFS		PFI		OS	
	n	MS	*p* *	MS	*p* *	MS	*p* *
**Integrin α2β1 high**	15	16	**0.035**	11	**0.034**	30	0.155
**Integrin α2β1 low**	27	29		25		45	
**Integrin α2β1 high/ERα high**	14	16	0.078	11	0.073	30	0.287
**Remaining combinations ^#^**	28	27		22		42	
**Integrin α2β1 high/PR high**	8	16	0.574	1119	0.578	27	0.526
**Remaining combinations ^#^**	34	24		19		42	
**Integrin α2β1 high/Her-2/neu +**	5	21	**0.043**	15	**0.037**	36	0.698
**Remaining combinations ^#^**	37	27		22		42	
**Integrin α2β1 high/EGFR high**	6	14	0.289	8	0.290	30	0.482
**Remaining combinations ^#^**	36	22		17		42	
**Integrin α2β1 high/HGFR high**	11	15	**0.004**	10	**0.004**	27	0.054
**Remaining combinations ^#^**	31	29		25		45	
**Integrin α2β1 high/IGFR high**	11	16	**0.045**	11	**0.043**	36	0.381
**Remaining combinations ^#^**	31	27		22		42	
**Integrin α2β1 high/MUC-1 high**	9	14	0.063	9	0.055	27	0.257
**Remaining combinations ^#^**	33	27		22		42	
**Integrin α2β1 high/CD44v6 high**	6	13	**0.000**	9	**0.001**	19	**0.025**
**Remaining combinations ^#^**	36	27		22		42	
**Integrin α2β1 high/Integrin αvβ3 high**	5	35	0.322	30	0.320	nr	0.162
**Remaining combinations ^#^**	37	22		17		42	

**Legend:** n: number of patients, MS: median survival (in months) in Kaplan–Meier estimator, *: *p*-value calculated by log-rank test. ^#^ The remaining combinations represent tumor samples which were integrin α2β1 high/biomarker X low, integrin α2β1 low/biomarker X high, or integrin α2β1 low/biomarker X low. nr: median survival not reached.

**Table 6 biomedicines-09-00289-t006:** Correlations between integrin α2β1 and the immune infiltrate.

Immune Infiltrate			Integrin α2β1		
		n	<20%	≥20%	*p* ^#^
**CD3 stromal**		48			**0.034**
	**Low**		13	14	
	**High**		17	4	
**CD3 intratumoral**		48			**0.017**
	**Low**		12	14	
	**High**		18	4	
**CD8 stromal**		48			0.133
	**Low**		14	13	
	**High**		16	5	
**CD8 intratumoral**		48			0.363
	**Low**		17	13	
	**High**		13	5	
**PD-1 stromal**		48			**0.049**
	**Low**		18	16	
	**High**		12	2	
**PD-1 intratumoral**		48			**0.002**
	**Low**		15	17	
	**High**		15	1	
**PD-L1 positivity**		48			**0.005**
	**No**		7	12	
	**Yes**		23	6	

**Legend:** n: number of patients, ^#^: *p*-value as calculated by Fisher’s exact two-tailed test.

## Data Availability

The data presented in this study are available on request from the corresponding author. The data are not publicly available due to protection of detailed patient-related data.

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
