# Peer review of "Integrin α2β1 Represents a Prognostic and Predictive Biomarker in Primary Ovarian Cancer"

_biomedicines, 2021, doi:10.3390/biomedicines9030289_

Round 1
Reviewer 1 Report
The authors have proposed Integrin α2β1 as a prognostic and predictive biomarker for OC. The manuscript is well organized and authors have tried to use mainly INC to evaluate their data. The specific comments are as follows:
- The introduction was not adequate as the background was not elaborately described.
- The authors have used a single method for validation of α2β1 . The sample size for this kind of study is small. Higher sample size will be more suitable for this study.
- The scoring of IHC was not clear and no representative images were included in this manuscript. The authors have talked about the distribution and staining patterns with out a
- As per recent US FDA and EMA the authors did not classify this marker. Please see following references.
US Food and Drug Administration–National Institutes of Health Biomarker Working Group. BEST (Biomarkers, EndpointS, and other Tools) Resource (FDA, 2016).
European Medicines Agency. Guideline on the clinical investigation of medicines for the treatment of Alzheimer’s disease (EMA, 2018)
FDA Center for Drug Evaluation and Research. Biomarker qualification: evidentiary framework (FDA, 2018).
https://www.nature.com/articles/s41582-020-0362-2/tables/1
- Keeping the status of the biomarker discovery field in mind, a single protein marker for staging any disease is not any more a valid idea. Need more validation in other biological specimens like blood and saliva along with tissue.
- I believe it would be more interesting to evaluate α2β1 positive tumors co-expressing HGFR or CD44v6 and evaluating them together would be more appealing to a broader audience.
Reviewer 2 Report
In this study, Dötzer et al. established Integrin α2β1 as the prognostic and predictive marker in primary ovarian cancer with the future potential to stratify patients for chemo- and immunotherapy and to design new targeted treatment strategies. I want to congratulate the authors for this work, and it will have a significant impact on ovarian cancer and the integrins field. The shortcoming of this study, as the authors mentioned themselves, is the small sample size. If authors want, they can think about my last point to confirm their results at the RNA level in bigger cohorts. Point 1 below is the major and needs to be addressed and the rest minor.
1. I really missed seeing the actual stainings. It would be great if the authors could show some staining examples, at least for the most important finding. e.g., low and high α2β1 and its correlation with EGFR and ERα, etc. The incorporation of staining examples will make the study look more 'reliable and assuring'. They can include staining panels in the supplementary if the main manuscript already has many tables and no space left. In my experience, this will also increase the citations for the article over time because the images would start popping up in various search engines' image search results.
2. The authors mentioned, "No impact of the extent of α2β1 expression on OS was observed". Any rational authors can think about this observation?
3. There are studies where other integrins that heterodimerize with β1 integrins, e.g., α5, show similar survival observation that the authors observed for α2β1. Can the authors discuss this in the discussion part?
4. Relationship, if any, between the TP53 status and differential α2β1 expression from this study would be interesting to know.
5. Do the authors want to comment on the intrapatient and interpatient tumor heterogeneity they observed during the course of this study?
6. Since the study had a low number of patients, as the authors acknowledged and to make this study even more impactful, authors can try looking for these correlations at the RNA level from the publically available ovarian cancer data sets, e.g, TCGA. Websites like KMplotter, Oncomine, and CBioportal can be used for this type of analysis and do not need any bioinformatics or programming experience. If similar correlations for the expression or survival are found, those can be put in the supplementary file. This analysis I mentioned is completely optional, and authors are free to include it or not in this paper.
The next step of the above strategy I mentioned will be to look for some enriched common mutations in the cancer-related genes with α2β1 high or low expression, but this is out of scope for this manuscript.
Reviewer 3 Report
In this manuscript Dötzer et al. have described integrin α2β1 as a prognostic and predictive marker in primary ovarian cancer with the potential to stratify patients for chemo- and immunotherapy and to design new targeted treatment strategies. The study population is not big, however is well defined. Indeed, authors emphasize that the main limitation of this study is the small cohort though it is representative and comparable to cohorts of other clinical trials.
The work is generally thorough and well done. However, I do have several concerns, as listed below.
- Integrins have very complex biology. At least in introduction a few basic statement on integrin biology are needed. It is important for readers to bear in mind the complexity of integrin pairing and promiscuity of integrin β1 which can form heterodimers with many different α subunits (subunit α2 is only one of them). Since integrin α2β1 is the central integrin heterodimer in this article it should be briefly introduced. Data are missing on the expression of this integrin in different tissues as well as expression in tumors. The ability of this integrin to bind extracellular matrix proteins is also important together with its role in tumor cell proliferation, migration, invasion and sensitivity to chemo and radiotherapy. This should be at least mentioned in introduction.
- High integrin α2β1 expression in primary ovarian cancer was found to be associated with an unfavorable prognosis. Patients with a high expression of integrin α2β1 showed a median PFS (progression free survival) of 16 months, which was significantly shorter compared to patients with a low α2β1 expression (29 months). In addition, high expression of the integrin α2β1 in primary ovarian cancer was found as an independent prognostic factor for a shorter PFI (platinum free interval). According to these data authors presume that integrin α2β1 might predict platinum-resistance in ovarian cancer. This is certainly an important proposition but should be discussed more extensively. At least some examples from the literature showing the connection between integrin expression and platinum resistance should be discussed.
- Low expression of integrin α2β1 could be a potential predictive marker for immunotherapy in ovarian cancer. This hypothesis should also be discussed further.
- I agree with authors that inhibition of integrin α2β1 should be considered as a possible target for ovarian cancer therapy. However, this statement should be critically discussed. Several inhibitors and antibodies have been developed and evaluated for integrin α2β1 inhibition in other tumor models. However, problems encountered in doing so should be discussed. Finally, obstacles in using integrins as therapeutic targets i.e. integrin crosstalk as well as integrin - growth factor crosstalk must be considered.
Round 2
Reviewer 1 Report
The authors have successfully addressed the issues raised in the review.